# Sphingosine-1-Phosphate Receptor Type 4 (S1P_4_) Is Differentially Regulated in Peritoneal B1 B Cells upon TLR4 Stimulation and Facilitates the Egress of Peritoneal B1a B Cells and Subsequent Accumulation of Splenic IRA B Cells under Inflammatory Conditions

**DOI:** 10.3390/ijms22073465

**Published:** 2021-03-27

**Authors:** Janik Riese, Alina Gromann, Felix Lührs, Annabel Kleinwort, Tobias Schulze

**Affiliations:** Experimental Surgical Research Laboratory, Department of General Surgery, Visceral, Thoracic and Vascular Surgery, Universitätsmedizin Greifswald, 17475 Greifswald, Germany; janik.riese@stud.uni-greifswald.de (J.R.); alina.gromann@stud.uni-greifswald.de (A.G.); andreasfelix.luehrs@stud.uni-greifswald.de (F.L.); annabel.kleinwort@med.uni-greifswald.de (A.K.)

**Keywords:** sphingosine-1-phosphate, peritoneal B cells, abdominal sepsis, lymphocyte trafficking, innate response activator B cells (IRA B cells)

## Abstract

Background: Gram-negative infections of the peritoneal cavity result in profound modifications of peritoneal B cell populations and induce the migration of peritoneal B cells to distant secondary lymphoid organs. However, mechanisms controlling the egress of peritoneal B cells from the peritoneal cavity and their subsequent trafficking remain incompletely understood. Sphingosine-1-phosphate (S1P)-mediated signaling controls migratory processes in numerous immune cells. The present work investigates the role of S1P-mediated signaling in peritoneal B cell trafficking under inflammatory conditions. Methods: Differential S1P receptor expression after peritoneal B cell activation was assessed semi‑quantitatively using RT-PCR in vitro. The functional implications of differential S1P_1_ and S1P_4_ expression were assessed by transwell migration in vitro, by adoptive peritoneal B cell transfer in a model of sterile lipopolysaccharide (LPS)‑induced peritonitis and in the polymicrobial colon ascendens stent peritonitis (CASP) model. Results: The two sphingosine-1-phosphate receptors (S1PRs) expressed in peritoneal B cell subsets S1P_1_ and S1P_4_ are differentially regulated upon stimulation with the TLR4 agonist LPS, but not upon PMA/ionomycin or B cell receptor (BCR) crosslinking. S1P_4_ deficiency affects both the trafficking of activated peritoneal B cells to secondary lymphoid organs and the positioning of these cells within the functional compartments of the targeted organ. S1P_4_ deficiency in LPS-activated peritoneal B cells results in significantly reduced numbers of splenic innate response activator B cells. Conclusions: The S1P-S1PR system is implicated in the trafficking of LPS-activated peritoneal B cells. Given the protective role of peritoneal B1a B cells in peritoneal sepsis, further experiments to investigate the impact of S1P_4_-mediated signaling on the severity and mortality of peritoneal sepsis are warranted.

## 1. Introduction

In 2017, an estimated 48.9 million cases of sepsis occurred worldwide [1]. After the respiratory system, the abdominal cavity (PerC) is the second most frequent site of septic infection [2]. In patients with abdominal sepsis, the most common pathogens isolated from the peritoneal cavity are Gram-negative bacteria as encountered in the intestinal microbiome [3]. The peritoneal cavity is populated with a wide variety of immune cells of the innate and adaptive immune system, including polymorphonuclear leukocytes, macrophages, B and T lymphocytes, dendritic cells, and resident mast cells [4]. Upon fecal contamination of the PerC, peritoneal cell populations undergo highly dynamic quantitative changes. While macrophage and B and T lymphocyte numbers decrease significantly during the initial hours of the infectious insult, neutrophils and monocytes exhibit increased numbers during the initial phase of abdominal sepsis [5]. Among the peritoneal lymphocyte populations, B1 B cells serve a pivotal role in the pathogenesis of abdominal sepsis and constitute approximately 35–70% of CD19^+^ B cells in the peritoneal cavity [6]. Notably, genetic deletion of these cells resulted in increased mortality in a murine model of sterile LPS-induced peritonitis [7]. LPS is a major constituent of the outer membrane of Gram-negative bacteria and can activate B cells via its interaction with the toll-like receptor 4 (TLR4) [8]. Upon exposure to LPS, peritoneal B1 B cells are stimulated to migrate to the spleen, where they produce higher amounts of natural IgM [9,10,11] and various cytokines [12,13]. Although the involvement of the CC and CXC families of chemokines in this process has been experimentally proven [14,15], the complexity of the signals regulating B1 B cell migration under septic conditions remains incompletely understood.

Sphingosine-1-phosphate (S1P) is a bioactive sphingolipid involved in regulating a plethora of biological processes (reviewed in [16,17]). Although S1P was initially considered mainly as an intracellular messenger [18,19], it has subsequently been shown that the majority of its biological functions, especially in the immune system, are regulated by its interaction with five membrane-bound, G-protein-coupled receptors (S1P_1_–S1P_5_) [20]. S1P binding to its membrane receptors regulates the trafficking of various immune cells and their positioning within the compartments of secondary lymphoid organs (SLOs), often in conjunction with the action of chemokines [17,21,22]. Our group has recently shown that peritoneal B cell subsets differentially express S1P receptor signatures under resting conditions, which results in differing migrational responses to S1P gradients [23]. Kunisawa et al. reported the implication of S1P signaling in peritoneal B cell trafficking and subsequent intestinal IgA production [24], while other authors suggested that the S1P system is not involved in the migratory processes of peritoneal B1 cells after LPS exposure [14].

S1P receptor expression is differentially regulated during the activation of various immune cells. These quantitative changes in S1PR expression result in alterations to the biological function of these cells, which primarily affect cell migration but also cytokine production [25,26]. The influence of peritoneal B cell activation on S1PR expression and the resulting functional consequences for pathological conditions involving strong peritoneal B cell activation (i.e., peritoneal sepsis) have not been investigated to date.

In the present paper, we demonstrate that the activation of peritoneal B cells with LPS—the main TLR agonist in Gram-negative abdominal sepsis [27,28]—results in the modification of their S1PR expression pattern and leads to profound functional alterations in their biological behavior during abdominal sepsis.

## 2. Results

### 2.1. S1P Receptor Expression Is Differentially Regulated during Peritoneal B Cell Activation In Vitro

During the activation of various immune cells, S1PR expression is differentially regulated [25,26]. To test whether peritoneal B cell activation results in changes in S1PR expression related to the function of the stimulating signal, S1PR expression was assessed at the mRNA level after peritoneal B cell activation with three different activation stimuli: (a) nonspecific stimulation with phorbol myristate acetate (PMA) and ionomycin, (b) crosslinking of the B cell receptor using agonistic anti-mouse IgM, and (c) specific TLR4 stimulation with LPS [8]. As previously shown, S1P_1_ and S1P_4_ were the two S1P receptor subtypes expressed in peritoneal B cells, while S1P_2_, S1P_3,_ and S1P_5_ were not detectable in all three subtypes of peritoneal B cells (B1a, B1b, and B2; resting or activated) at a level greater than the background signal. Although stimulation with PMA and with anti-mouse IgM did not result in any significant change in S1PR expression, TLR4 stimulation with LPS led to the significant downregulation of S1P_1_ expression in all subtypes of peritoneal B cells. In contrast, S1P_4_ expression was only downregulated in peritoneal B1a and B1b, but not in B2 B cells after TLR4 stimulation with LPS (Figure 1).

### 2.2. Peritoneal B Cell Activation via TLR4 Leads to Reduced Chemotactic Response to S1P In Vitro

S1P has been shown to regulate peritoneal B cell migration via S1P_1_ and S1P_4_ [23]. To assess the functional consequences of activation-induced modification of S1PR expression on peritoneal B cells, the migration of these cells to an S1P gradient was tested in vitro. Since differential expression of S1P_4_ was observed in the peritoneal B cell subsets, these experiments were performed using wild type (*wt*) and S1P_4_-deficient (*s1pr_4_^−/−^*) mice. There was a significant reduction in S1P‑induced chemotactic response in *wt* B1a and B1b cells after previous LPS‑activation, while peritoneal B2 B cell migration remained unchanged before and after activation (Figure 2). In *s1pr_4_^−/−^* peritoneal B1a cells, migration to the S1P gradient was significantly lower than in *wt* B1a B cells, both with and without LPS‑induced activation. Finally, no differences in chemotactic behavior to S1P were observed between *wt* and *s1pr_4_^−/−^* cells could be observed in peritoneal B1b and B2 B cells at resting conditions and after LPS stimulation.

### 2.3. Impact of S1P_4_ Deficiency on the Trafficking of Adoptively Transferred Peritoneal B Cells in Sterile LPS-Induced Peritonitis

After demonstrating that S1P_4_ contributes to the chemotactic response of peritoneal B1a B cells to S1P under both nonactivating and activating conditions, we studied the contribution of S1P_4_‑mediated S1P signaling to peritoneal B cell trafficking in a murine model of LPS‑induced peritonitis. To achieve this, CSFE‑labeled *wt* or *s1pr_4_^−/−^* peritoneal cells were transferred into the PerC of *wt* mice. After 48 h of repeated intraperitoneal LPS‑stimulation, animals were sacrificed 72 h after the start of the experiment, and in various organs the number of labeled peritoneal B cells in various tissues was measured by FACS. In the PerC, the percentage of transferred B1a B cells present 72 h after transfer was lower in the *wt* than the *s1pr_4_^−/−^* genotype (3.5 versus 1.6% for *s1pr_4_^−/−^* and *wt* cells, respectively). In contrast, there were no differences in the percentages of transferred *wt* B1b and B2 B cells to that of *s1pr_4_^−/−^* cells (Figure 3A). In the omentum, which is one of the main exit routes of peritoneal B cells, a comparable percentage of transferred peritoneal B1b and B2 B cells were detected 72 h after transfer. In contrast, a highly reduced proportion of transferred *s1pr_4_^−/−^* B1a B cells compared to *wt* B1a B cells was observed in the omentum (Figure 3B). No significant differences were observed between the percentages of transferred peritoneal B cells of all three lineages between the *s1pr_4_^−/−^* genotype and the *wt* in parathymic lymph nodes (ptLN) (Figure 3C). The ratio of transferred *s1pr_4_^−/−^* B1a B cells to *wt* B1a B cells was significantly reduced in the spleen and mesenteric lymph nodes (mLN) (Figure 3D,E). Moreover, no differences between *s1pr_4_^−/−^* B1b cells and *wt* B1b cells, as well as *s1pr_4_^−/−^* B2 cells and *wt* B2 cells, were observed in these two organs.

### 2.4. Impact of S1P_4_ on the Localization of Migrating Peritoneal B Cells within the Spleen, Lymph Nodes, and Peyer’s Patches after LPS Stimulation

Within SLOs, S1P_4_ deficiency induced characteristic alterations to the distribution of migrating peritoneal B cells within the structural compartments of these organs. While the *wt* B cells were loosely scattered over the red pulp or localized within the follicles, *s1pr_4_^−/−^* B cells showed a distinctive distribution. Rarely located within the follicles, *s1pr_4_^−/−^* B cells were predominantly arranged in ring-shaped figures around the follicles. Quantification of these observations revealed that *s1pr_4_^−/−^* B cells were clustered within the marginal zone, while they showed a lower frequency in other compartments of the spleen (Figure 4A–C). Likewise, a perifollicular clustering of transferred *s1pr_4_^−/−^* peritoneal B cells could be observed in the mesenterial and parathymic lymph nodes (Appendix A).

In the Peyer’s patches (PPs), transferred peritoneal B cells of both genotypes localized in great numbers at the outer edge of the PPs. However, while *wt* B cells were also mostly present at the luminal surface of the PPs below the subepithelial dome (SED), almost no *s1pr_4_^−/−^* B cells could be found at this location. Instead, numerous *s1pr_4_^−/−^* B cells were located on the serosal side of the PPs. Within the adjacent villi, reduced numbers of *s1pr_4_^−/−^* peritoneal B cells were present when compared to *wt* peritoneal B cells. (Figure 5A–G).

### 2.5. Impact of S1P_4_ Expression on the Number of Splenic IRA B Cells after Peritoneal LPS Stimulation

In murine models of abdominal sepsis, a small fraction of peritoneal B1a cells has been shown to migrate to the spleen, where they form GM-CSF expressing innate response activator (IRA) B cells and influence the outcome of sepsis [12]. In the model of sterile LPS‑induced peritonitis, there were significantly fewer labeled GM-CSF-positive *s1pr_4_^−/−^* IRA B cells than *wt* IRA B cells found in the spleen. These differences concerned both the population of IRA B cells in the red pulp and the smaller fraction of IRA B cells in the white pulp (Figure 6A–C).

### 2.6. Influence of S1P_4_ Expression on the Composition of Peritoneal B Cell Populations and Cytokine Levels in A Murine Model of Polymicrobial Abdominal Sepsis

After the induction of abdominal polymicrobial sepsis, the composition of peritoneal cell populations undergoes profound remodeling [5]. To investigate the role of S1P_4_ in B cell dynamics after a polymicrobial challenge of the PerC, the colon ascendens stent peritonitis (CASP) model was used. In contrast to sterile LPS-induced peritonitis, peritoneal B1a B cell numbers were significantly reduced in *s1pr_4_^−/−^* mice when compared to the *wt* control animals both 24 h and 7 days after sepsis induction. No differences concerning peritoneal B1b and B2 B cells were observed between *s1pr_4_^−/−^* mice and *wt* mice at these time points (Figure 7A–C).

In order to assess the potential functional consequences of these observations, the following cytokines were quantified in peritoneal lavage fluid, plasma, and spleen: IL-1, -2, -3, -6, -10, -12p70, -17, CXCL13, GM-CSF, IFN-γ, MCP-1, TGF-β, and TNF-α. Both lavage and plasma showed a significantly higher IL-10 concentration in *s1pr_4_^−/−^* mice when compared to *wt* animals. Moreover, IL-6 levels in the plasma of *s1pr_4_^−/−^* animals were slightly higher when compared to the *wt* controls. In contrast, IL-6 levels in the peritoneal lavage and the spleen were similar in *s1pr_4_^−/−^* and *wt* animals (Figure 7D–F). A complete list of measured cytokine concentrations can be found in the Appendix A.

### 2.7. S1P_4_ Deficiency Results in a Significant Reduction of IRA B Cells in the Spleens of CASP Mice

As observed in the model of sterile LPS‑induced peritonitis, there were significantly fewer IRA B cells in the red pulp of *s1pr_4_^−/−^* animals when compared to *wt* mice seven days after CASP induction. This difference was due to a significant reduction of IRA B cells localized within the red pulp, while the small fraction of IRA B cells localized in the white pulp showed no differences between *s1pr_4_^−/−^* and *wt* animals (Figure 8A,B). However, while IRA B cells in *wt* animals were loosely scattered over the white pulp, *s1pr_4_^−/−^* cells showed a preferred localization in the marginal zone (Figure 8C,D).

## 3. Discussion

Lymphocyte trafficking through central and peripheral lymphoid organs, as well as through peripheral tissues and body cavities, is indispensable for the maintenance of immune homeostasis and the induction of an effective immune response after an infectious challenge [29,30]. While the molecular events governing the passage of lymphocytes from the blood compartment into the lymph node and the exit of lymphocytes from the lymph node have been studied in detail, there remains considerable scientific debate on the mechanisms involved in the exit of lymphocytes from body cavities and the subsequent localization of these cells in SLOs [31]. The results of the present study demonstrate that the S1P system regulates lymphocyte exit from the peritoneal cavity under inflammatory conditions. Furthermore, we show that the S1P system influences the subsequent localization of these cells within the SLOs.

In the present work, we confirm previous results on S1PR expression showing that both S1P_1_ and S1P_4_ are expressed in peritoneal B cell subsets, while S1P_2_, S1P_3,_ and S1P_5_ mRNA could not be detected in our model [23]. Both S1P_1_ and S1P_4_ were differentially regulated upon stimulation with LPS via the TLR4 receptor, while stimulation with PMA or B cell receptor (BCR) crosslinking with anti-IgM did not result in a significant modification of S1P_1_ or S1P_4_ expression levels. It has previously been shown that S1PR expression depends on the activation status of various immune cells. Müller et al. reported that macrophage activation and polarization results in the modification of S1P_1_ and S1P_4_ expression in vitro [26]. Similarly, Gräler et al. described the downregulation of both S1P_1_ and S1P_4_ on anti-CD3- and anti-CD28-stimulated CD4^+^ and CD8^+^ T cells in vitro [25]. In vivo, the downregulation of S1P_1_ expression upon specific stimulation of DO11.10 transgenic T cells has been shown to be implicated in T cell retention within SLOs [32]. Cinamon et al. showed that splenic marginal zone B cells downregulated S1P_1_, S1P_3,_ and S1P_4_ after stimulation with LPS or specific antigens [20]. Finally, Gohda et al. reported downregulation of S1P_1_ expression in activated B cells isolated from Peyer’s patches both in vivo and in vitro [33]. Previous reports on the regulation of S1P expression in peritoneal B cells show partially conflicting results. Ha et al. noted the substantial downregulation of S1P_1_ after TLR stimulation, while Moon et al. observed no downregulation of S1P_1_ expression after exposure to LPS—a divergence that is likely due to sensitivity differences in the techniques used for receptor detection [9,14]. Importantly, this is the first report on activation-induced differential expression of the second S1PR expressed in peritoneal B cells (S1P_4_). While the downregulation of S1P_1_ occurred in all three peritoneal B cell subsets upon TLR stimulation, the downregulation of S1P_4_ was limited to B1a and B1b B cells, which indicates the specific regulation of S1P_4_ expression during the activation of different peritoneal B cell subsets. Interestingly, S1PR regulation was also dependent on the stimulus used for B cell activation in vitro. In contrast to TLR4-mediated stimulation, the unspecific activation of peritoneal B cells using the protein kinase C activator phorbol myristate acetate and calcium ionophore ionomycin did not result in differential S1PR receptor expression in B1 or B2 B cells. Similarly, the stimulation of peritoneal B cells by crosslinking their BCR with anti-IgM did not result in detectable changes in the expression levels of S1P_1_ or S1P_4_. Since BCR crosslinking is a very weak activator of B1 B cells [34], the lack of effect of IgM treatment on S1PR expression levels may be a consequence of the weak stimulation of these cells. However, in B2 B cells, BCR crosslinking results in efficient activation [34]. Nevertheless, no downregulation of S1P_1_ or S1P_4_ occurred in these cells after anti-IgM treatment. These findings indicate that the modification of S1PR expression upon B cell stimulation is specific for both the peritoneal B cell subset and the stimulus used for activation.

To assess the functional consequences of these subtype-specific and activation-induced changes of S1PR receptor expression for the chemotactic response of peritoneal B cells to S1P, in vitro migration to an S1P gradient was assessed using a transwell chemotaxis assay. While Ha et al. already demonstrated that downregulation of S1P_1_ results in reduced migration of the total B1 B cell fraction to an S1P gradient in vitro, we used *s1pr_4_^−/−^* peritoneal B cells to distinguish between the effects mediated by S1P_1_ and S1P_4_ [9]. As expected, activated *wt* B1 B cells showed a significantly lower chemotactic response than *wt* B1 B cells. In resting B1 B cells, S1P_1_ and S1P_4_ had a synergistic effect on S1P-induced chemotaxis since resting *s1pr_4_^−/−^* B1 B cells showed reduced S1P‑induced chemotaxis. Notably, these results are consistent with a previous study [23]. Following activation, this synergistic effect persisted since activated *wt* B1a B cells continued to show increased migration to S1P when compared to activated *s1pr_4_^−/−^* B1a B cells. In contrast, while resting *wt* B1b B cells showed a chemotactic response to S1P that was mediated additively by both S1P_1_ and S1P_4_, there was no difference in the chemotactic response to S1P between activated *wt* and *s1pr_4_^−/−^* B1b B cells, which indicates that S1P_4_ does not participate in the elicitation of a chemotactic response to S1P in these cells under activated conditions. Finally, the chemotactic response to S1P was lower in B2 B cells than in B1 B cell subsets, and neither the activation nor the lack of S1P_4_ resulted in a statistically significant reduction in S1P‑induced chemotaxis. These observations indicate that the activation-induced downregulation of S1P_1_ and S1P_4_ resulted in a substantial decrease in chemotaxis to an S1P gradient in B1 cells. While S1P_1_ and S1P_4_ are involved in peritoneal B1a cells, our results indicate that S1P_4_‑mediated signaling has no additional effect on S1P‑induced chemotaxis in B1b cells.

Our data indicate that S1P_4_ differentially affects the migratory behavior of resting and activated peritoneal B cells in vitro. To evaluate the functional role of S1P_4_-mediated signaling for peritoneal B cell migration in vivo, the trafficking of TLR‑activated peritoneal B cells was further investigated in a combined model of adoptive peritoneal transfer and LPS‑induced sterile peritonitis. Under homeostatic conditions, peritoneal B cells egress from the peritoneal cavity primarily via the omentum and/or the parathymic lymph nodes [35]. In our experimental setting, increased numbers of peritoneal *s1pr_4_^−/−^* B1a B cells indicate a reduced exit of *s1pr_4_^−/−^* B1a B cells from the peritoneal cavity. While *s1pr_4_^−/−^* B1a B cell numbers were also significantly reduced in the omentum, they showed normal numbers in the parathymic lymph nodes. The synthesis of these findings implies that a lack of S1P_4_ expression interfered with the exit route involving the omentum, which increased the peritoneal retention of *s1pr_4_^−/−^* B1a B cells. The reduction in B1a B cells exiting the peritoneal cavity further resulted in the significantly reduced migration of transferred B1a B cells to the spleen. This observation contradicts a report by Kunisawa et al. showing that the nonspecific blockage of S1P signaling with FTY720 induced decreased peritoneal B cell numbers within the peritoneal cavity and omentum, while their number increased in parathymic lymph nodes [24]. Due to substantial differences in the experimental setting, direct comparisons between these contradictory results are problematic. For example, while Kunisawa et al. investigated unspecific blockage under homeostatic conditions in SCID mice, the present study assessed the specific influence of the S1P_4_ receptor after the TLR4‑mediated activation of peritoneal B cells. Given the experimental differences, both reports reveal the role of S1P signaling in the exit of peritoneal B cells from the peritoneal cavity. Moreover, our results highlight the involvement of S1P_4_‑mediated signaling in the regulation of peritoneal B1a B cell exit via the omental route as well as the peritoneal exit of activated B1b and B2 B cells being unaffected by S1P_4_ deficiency.

In contrast to observations made in the model of sterile peritonitis, where transferred peritoneal *s1pr_4_^−/−^* B1a B cells showed increased peritoneal cell numbers 72 h after i.p. LPS injection, B1a B cell numbers were lower in *s1pr_4_^−/−^* animals compared to *wt* animals both 24 h and 7 days after peritonitis induction. This corresponds to previous reports, which noted reduced B1a B cell numbers in *s1pr_4_^−/−^* animals under homeostatic conditions [23]. The reduced exit capacity of *s1pr_4_^−/−^* B1a B cells demonstrated in the transfer model of sterile peritonitis may be insufficient to compensate for the reduced pre-existing B1a B cell numbers in *s1pr_4_^−/−^* animals in the CASP model. Moreover, peritoneal B cell numbers result from exit and entry processes from and into the peritoneal cavity, respectively. S1P‑mediated signaling has been shown to influence both outbound and inbound trafficking of peritoneal B cells [24]. While the transfer experiments of *s1pr_4_^−/−^* peritoneal B cells predominantly reflect outbound trafficking, the CASP model used in the present study assesses both arms of peritoneal B cell trafficking, which suggests that a lack of S1P_4_ might also interfere with B1a B cell entry to the peritoneal cavity. Finally, in contrast to the transfer model of sterile peritonitis in *wt* animals used in the present study, altered cellular interactions between *s1pr_4_^−/−^* peritoneal B cells and other *s1pr_4_^−/−^* cell types cannot be excluded in the CASP model performed using *s1pr_4_^−/−^* animals [26,36]. Thus, further research is required to determine the relative importance of these three potential mechanisms for inducing the differences in peritoneal B cell subpopulations observed in the CASP model.

B1 B cell deficiency results in increased mortality in various animal models of abdominal sepsis [7,37]. In an LPS‑induced abdominal sepsis model, B1 B cells have been shown to attenuate the plasma and tissue levels of proinflammatory cytokines (TNF‑α and IL-6) produced by macrophages via an IL-10‑dependent mechanism [7]. In a polymicrobial sepsis model, the protective effect of B1a B cells was also mediated by IL-10 produced by B1a B cells [37]. In the CASP model, S1P_4_ deficiency resulted in significantly increased IL-10 levels in both the plasma and peritoneal lavage fluid. Thus, further experiments are required to determine whether *s1pr_4_^−/−^* cells intrinsically produce higher IL-6 or whether the observed increase in IL-6 levels results from an aberrant interaction of *s1pr_4_^−/−^* B1a B cells with other IL-6 producing cells.

A second protective mechanism involving peritoneal B1a B cells—in both abdominal sepsis and pneumonia—is the generation of IRA B cells that are differentiated from the peritoneal B1a B cell population and represent 2–4% of the splenic B cells 4 days after LPS administration [12]. These cells exert their protective role via the increased generation of polyreactive IgM under the control of enhanced autocrine GM-CSF secretion [38]. In accordance with the finding that S1P_4_ deficiency impedes B1a B cell exit from the peritoneal cavity, reduced IRA B cell numbers in the spleen were observed in both sepsis models. These findings demonstrate that S1P_4_ is implicated in controlling B1a B cell trafficking from the peritoneal cavity to the spleen and IRA B cell formation in this organ.

Other than the quantitative control of peritoneal B cell trafficking to SLOs, our findings also support a regulatory role of S1P_4_ in the positioning of migrating peritoneal B cells within the different compartments of these lymphoid organs. In splenic white pulp, transferred *wt* peritoneal B cells preferentially localized within the follicles while *s1pr_4_^−/−^* B cells localized outside of follicular structures. B cell entry into the follicles is controlled by CXCR5−CXCL13 interaction and can be inhibited by pertussis toxin, which suggests the involvement of a G-protein-coupled receptor [39]. In marginal zone splenic B cells, the balance between S1P_1_, S1P_3,_ and CXCR5 defines the localization of these cells within the marginal zone and the follicle [21,40]. Our results suggest that S1P_4_ deficiency results in the insufficient entry of migrating peritoneal B cells in splenic follicles. Likewise, migrating peritoneal B cells were predominantly clustered around the follicles in mesenteric lymph nodes, thereby reinforcing the hypothesis that S1P_4_ deficiency results in the blocking of activated peritoneal B cell entry into the follicles of SLOs. In PPs, the localization of *s1pr_4_^−/−^* peritoneal B cells was similar to that of IgA^+^ B cells in FTY720 treated mice, which indicates an immigration blockade from the lymph vessels around the basal side of the PPs [33].

Our in vivo results convincingly prove the functional significance of LPS-induced S1P receptor downregulation for peritoneal B cell migrational capacity to an S1P gradient. The in vitro data indicate that despite being downregulated to lower expression levels, S1P_4_ maintains a regulatory effect on peritoneal B cell migration under septic conditions. This observation confirms the principle that downregulated chemotactic receptors may maintain functional competency [41].

In conclusion, the experimental results presented herein show that the S1P-S1PR system is implicated in the migration of TLR4-activated peritoneal B cells. The two S1PRs expressed in peritoneal B cell subsets (i.e., S1P_1_ and S1P_4_) are differentially regulated upon stimulation with the TLR4-agonist LPS, but not with PMA/ionomycin or BCR crosslinking. S1P_4_ deficiency affects both the trafficking of activated peritoneal B cells to SLOs and the positioning of these cells within the functional compartments of the target organ. Finally, S1P_4_ deficiency in TLR4-activated peritoneal B cells results in significantly lower numbers of splenic IRA B cells. Further experiments to investigate the impact of S1P_4_‑mediated signaling on the severity and mortality of peritoneal sepsis are required.

## 4. Materials and Methods

### 4.1. Mice

Female C57BL/6J mice were purchased from Charles River (Sulzfeld, Germany) and kept for at least 2 weeks before the initiation of experiments to adapt to local conditions. *s1pr_4_^−/−^* mice on a C57BL/6J background were bred under specific pathogen-free conditions in the Zentrale Service- und Forschungseinrichtung für Versuchstierkunde (ZSFV, Greifswald, Germany). Mice aged between 10 and 14 weeks were used for all experiments. All animal care practices and experimental procedures were performed in accordance with the German Animal Protection Law (*TierSchuG*) and controlled by the veterinary government authority (Landesamt für Landwirtschaft, Lebensmittelsicherheit und Fischerei Mecklenburg-Vorpommern; LALLF-MV; Project identification codes: 7221.3-1-036/17-1 and 7221.3-1.1-044/19).

### 4.2. Cell Isolation from the Peritoneal Cavity

Murine peritoneal cavity cells were obtained by peritoneal lavage. Peritoneal lavage was performed via the intraperitoneal injection (i.p.) of 10 mL ice-cold PBS supplemented with either 2% fetal calf serum (Biochrom, Berlin, Germany) for RNA isolation and adoptive cell transfer or 0.5% fatty-acid-free bovine serum albumin (BSA, Sigma-Aldrich, St. Louis, MO, USA) for cell culture experiments involving the subsequent use of S1P. Absolute cell numbers were determined using BD Trucount tubes (BD Biosciences, Franklin Lakes, NJ, USA).

### 4.3. Flow Cytometry

Nonspecific binding was blocked with an anti-FcII/III antibody (anti-CD16/32; BD Pharmingen, Heidelberg, Germany). The following antibodies and conjugates were used in the experiments in appropriate combinations: anti-CD5-PE/Cy7 (clone 53-7.3; Biolegend, San Diego, CA, USA), anti-CD19-eFluor 660 (clone 1D3; eBioscience, Thermo Fisher Scientific, Waltham, MA, USA), anti-CD23-PE (clone B3B4; eBioscience), anti-CD3-FITC (clone 145-2c11; Biolegend), anti-CD11b-APC-eFluor 780 (clone M1/70; eBioscience), anti-IgD-V450 (clone 11-26c.2a; BD Biosciences), and anti-IgM-BV650 (clone R6-60.2; BD Biosciences). For dead cell exclusion, cells were stained with 7-AAD Viability Staining Solution (Biolegend) before analysis. Labeled cells of the transfer experiment were stained with the following antibodies: anti-CD11b-APC (M1/70; eBioscience), anti-CD19-eFluor 660 (clone 1D3; eBioscience), anti-CD23-PE (B3B4; eBioscience), anti-CD3-BV605 (145-2C11; Biolegend) and anti-CD5-PE/Cy7 (53-7.3; Biolegend). Stained cells were analyzed using a BD LSR II Flow Cytometer (BD Biosciences) and evaluated with FlowJo software (Version 10, LLC, Ashland, OR, USA). Fluorescence-activated cell sorting was performed using a MoFlo Astrios-EQ (Beckman Coulter, Brea, CA, USA) cell sorter and gated in Summit 6.3 (ROHS by Beckman Coulter). Detailed information on the antibodies used in our experiments is summarized in Appendix A. B cell subpopulations were identified using the following antibody panels: B1a B cells: CD19^+^CD5^+^CD23^−^IgM^+^IgD^−^; B1b B cells: CD19^+^CD5^−^CD23^−^IgM^+^IgD^−^; B2 B cells: CD19^+^CD5^−^CD23^+^IgM^low^IgD^+^.

### 4.4. In Vitro Activation of Peritoneal B Cell Subpopulations

After the isolation of peritoneal cells by peritoneal lavage, equal numbers of cells were incubated with either 10 µg/mL LPS (O111:B4, E. coli; Sigma-Aldrich, St. Louis, MO, USA), 10 µg/mL anti-mouse Igµ chain (Thermo Fisher Scientific, Waltham, MA, USA), or 1 ng/mL phorbol myristate acetate and 1 µg/mL ionomycin (both Sigma-Aldrich) for 24 h for activation. All cultures took place in an RPMI-1640 supplemented with 10% charcoal-adsorbed FCS, penicillin/streptomycin (100 U/mL), and 2-mercaptoethanol (2 mM) at 37 °C and 5% CO_2_.

### 4.5. RT-PCR of S1P Receptors in Peritoneal B Cells after In Vitro Stimulation

FACS‑sorted peritoneal B1a, B1b, and B2 B cells were loaded onto a QIAshredder spin column (QIAGEN, Venlo, NL, USA) and homogenized. For RNA isolation, the RNeasy Mini Kit (QIAGEN) was used according to the manufacturer’s instructions. RNA concentrations were measured using a NanoDrop (Thermo Fisher Scientific). The purity of isolated RNA was evaluated using an Agilent RNA 6000 Pico Kit (Agilent Technologies Deutschland GmbH, Waldbronn, Germany). For cDNA synthesis, identical amounts of RNA (30 ng) were reverse transcribed using the QuantiTect Reverse Transcription Kit (QIAGEN) according to the manufacturer’s recommendations. Real-time RT‑PCR of S1P_1–5_ receptors and β_2_-microglobulin expression was performed using the QuantiTect SYBR Green PCR Kit (QIAGEN) and an ABI Prism 7300 Sequence Detection System (Applied Biosystems, Foster City, CA, USA). The analysis was performed manually. Semi-quantitative gene expression was calculated according to the 2^−ΔΔCt^-method normalized to β_2_-microglobulin. Primers were designed using Primer3 software [42] and synthesized by BIOTEZ (Berlin, Germany). Primer sequences are listed in Appendix A. Statistical analysis was performed using mean 2^−ΔΔCt^‑values from biological replicates according to [43]. The fold change (FC) of medium control for each sample was computed as follows: $FC=\frac{-2^{\Delta C_{T}}}{\mu_{Medium}}$.

### 4.6. Transwell Migration Assay of Peritoneal B Cells after LPS Stimulation

Following the isolation of peritoneal cells, samples were incubated in migration medium (RPMI-1640 + GlutaMAX-I (Gibco, Thermo Fisher Scientific, Waltham, MA, USA), 0.5% fatty-acid free BSA (Sigma-Aldrich, St. Louis, MO, USA), 25mM HEPES (Biochrom, Holliston, MA, USA), 100 U/mL penicillin with 100 µg/mL streptomycin (Gibco) or migration medium with 10 µg/mL LPS for 24 h at 37 °C and 5% CO_2_. Subsequently, 0.5 × 10^6^ cells in 100 µL medium were loaded into the upper chamber of a transwell insert (5 µm pore size; Corning Costar, Cambridge, MA, USA). The lower chambers were filled with 600 μL of migration medium either with or without 10 nM S1P. For three wells per condition, cells were directly added to the lower well without using the insert. The numbers of B1a, B1b, and B2 B cells migrating to the lower chamber were determined by FACS after staining with the aforementioned antibody panels and using BD Trucount tubes (BD Biosciences). Total migration was calculated as follows: (absolute cell number of the lower chamber with S1P/absolute cell number in the lower chamber without insert) × 100. Nonspecific migration was calculated as follows: (absolute cell number of the lower chamber without S1P/absolute cell number in the lower chamber without insert) × 100. Finally, specific migration (%) was calculated by subtracting the nonspecific migration from the total migration.

### 4.7. Adoptive Cell Transfer under LPS Stimulation

Peritoneal cells from C57BL/6J (*wt*) and *s1pr_4_^−/−^* mice were labeled with CFSE (Biolegend) according to the manufacturer’s instructions. Thereafter, 5 × 10^6^ cells from either *wt* or *s1pr_4_^−/−^* mice were transferred to the peritoneal cavity of C57BL/6J mice (*wt*) by i.p. injection. Simultaneously, intraperitoneal LPS‑stimulation was performed by injecting 10 µg LPS in 500 µL of isotonic NaCl solution intraperitoneally. After 24 and 48 h, the mice again received 10 µg of LPS in 500 µL of isotonic NaCl solution intraperitoneally. Finally, mice were euthanized after 72 h by cervical dislocation and their organs were harvested.

### 4.8. Colon Ascendens Stent Peritonitis (CASP)

CASP surgery was performed as previously described [44]. Briefly, mice were anaesthetized and placed in a supine position. A 1 cm midline incision of the skin was performed and the abdominal muscles and the peritoneum along the linea alba were opened. After identifying the colon ascendens and the cecum, a plastic stent (18 G needle from BD Bioscience, Heidelberg, Germany) was inserted into the antimesenteric wall of the cecum 1.0 cm distal to the ileocecal valve (7/0 suture). Stool was milked out and fluid resuscitation was performed via intraperitoneal administration with 0.5 mL of fluid solution. The peritoneum and skin were then closed (two-layer continuous, 5/0 suture).

Twenty-four hours or seven days after sepsis induction, mice were anaesthetized and then sacrificed by cervical dislocation to allow organ harvesting. Tissue samples for microscopic analysis were embedded in TissueTek (Sakura Finetek Europe B.V., Alphen aan den Rijn, NL) immediately after organ removal and then snap-frozen in isopentane cooled by liquid nitrogen. All samples were stored at −80 °C until further use. Organ samples for flow cytometric analysis were processed immediatly after organ removal.

### 4.9. Cytokine Analysis

Cytokine concentrations in the peritoneal lavage fluid, blood plasma, and spleen were determined using a Custom Mouse Panel LEGENDplex Multi-Analyte Flow Assay Kit (Biolegend) according to the manufacturer’s instructions. The following substances were quantified: IL-1β, -2, -3, -6, -10, -12p70, -17A, CXCL13, GM-CSF, IFN-γ, MCP-1, TGF-β, and TNF-α.

### 4.10. Fluorescence Microscopy of Splenic Tissue Sections

Cryostat sections (3 µm) were dried for 24 h and then fixed in acetone at −20 °C for 10 min. Nonspecific binding sites were blocked with PBS + 10% FCS for 30 min. Biotin-binding sites were blocked with the Dako Biotin Blocking System (Dako North America Inc., Carpinteria, CA, USA) according to the manufacturer’s instructions. Sections were stained specific antibodies at 4 °C for 16 h. Sections were then incubated with streptavidin-rhodamine at room temperature for 1 h and cell nuclei were stained with Draq5 (Biolegend) or DAPI (Molecular Probes, Eugene, OR, USA). Stained sections were stored at 4 °C for at least 24 h before microscopy. All images for an experiment were acquired on the same day at a constant exposure time and magnification. The downstream segmentation and classification of cells were performed using QuPath [45]. B cells were identified as B220^+^ and IRA B cells as B220^+^GM-CSF^+^IgM^+^. An example for IRA B cell identification is shown in Appendix A. Fluorescence intensity of CFSE‑labeled cells was enhanced using polyclonal anti-FITC-Alexa488. Transferred B cells were identified as B220^+^ CFSE^+^.

### 4.11. Statistical Analysis

Statistical tests were performed using GraphPad Prism Software (Version 6.01, GraphPad Software, Inc., La Jolla, CA, USA). Groups were tested for Gaussian distribution using the Shapiro−Wilk test. When groups were normally distributed, statistical analysis was performed by either a *t*-test or one-way analysis of variance (ANOVA) and Dunnett’s multiple comparison test; otherwise, either Mann−Whitney-U test or Kruskal−Wallis test with Dunn’s multiple comparison test was used. *p*-values < 0.05 were considered to be significant. Significance in the graphs is labeled as follows: * *p* < 0.05, ** *p* < 0.01, or *** *p* < 0.001.

## Figures and Tables

**Figure 1 ijms-22-03465-f001:**
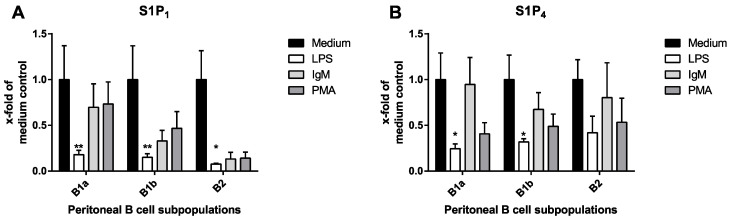
Expression of sphingosine-1-phosphate (S1P) receptor subtypes S1P_1–5_ in peritoneal B cell subpopulations. After in vitro stimulation, sphingosine-1-phosphate receptor expression in peritoneal B cell subtypes was determined by quantitative polymerase chain reaction (qPCR). Gene expression was normalized to β_2_-microglobulin. Fold change in the mRNA expression of medium control for S1P_1_ (**A**) and S1P_4_ (**B**) receptor. *p*‑values were computed based on the medium control from 2^−ΔΔCt^‑values using a Kruskal−Wallis test with Dunn’s multiple comparisons test. * *p* < 0.05, ** *p* < 0.01.

**Figure 2 ijms-22-03465-f002:**
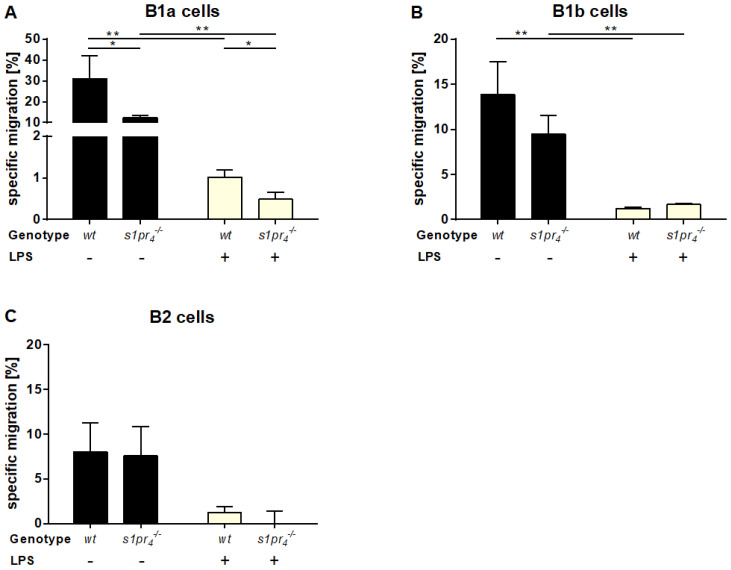
In vitro migration of peritoneal B cell subpopulations. The chemotactic response of peritoneal B1a (**A**), B1b (**B**), and B2 (**C**) B cell subpopulations to an S1P gradient with and without LPS-induced activation was analyzed in vitro in a transwell migration assay. Values represent the mean (+SEM) of *n* = 6 animals per condition. * *p* < 0.05, ** *p* < 0.01.

**Figure 3 ijms-22-03465-f003:**
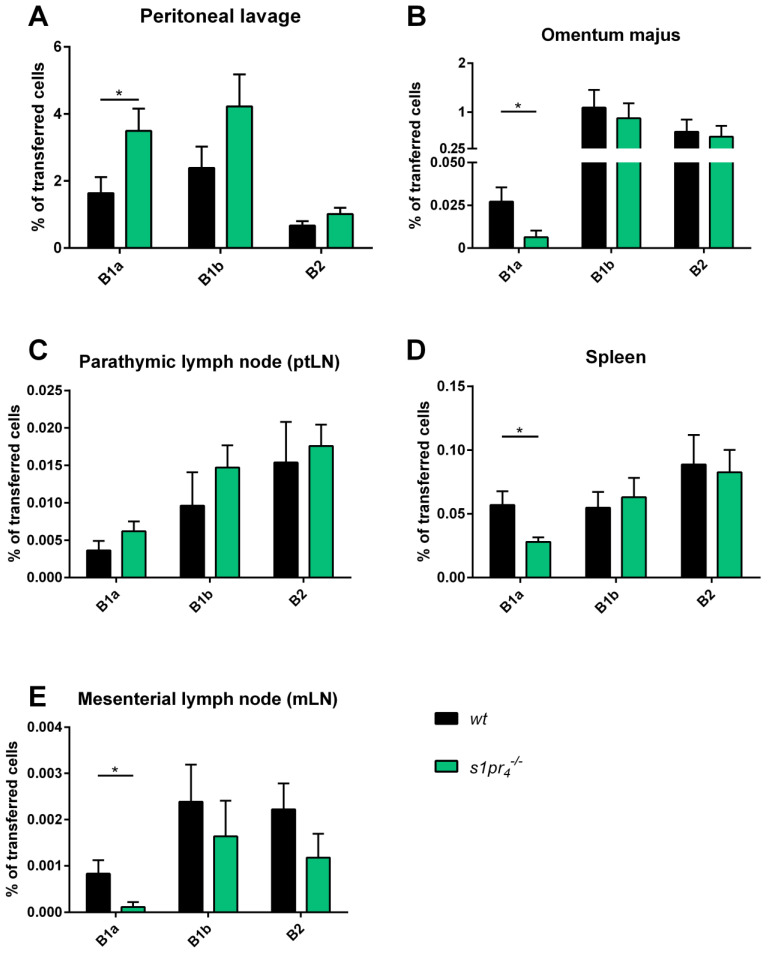
In vivo migration of intraperitoneal transferred peritoneal B cells after LPS stimulation. After the adoptive intraperitoneal transfer of labeled *wt* or *s1pr_4_^−/−^* peritoneal B cells into *wt* mice and 72 h of pulsed LPS treatment, the number of transferred cells was determined in the peritoneal cavity (**A**), the putative exit routes omentum (**B**), and parathymic lymph node (**C**), and the spleen (**D**) and mesenterial lymph node (**E**). Bars depict the mean values (+SEM) of total cell numbers for peritoneal B cell populations per organ (*n* = 10). * *p* < 0.05.

**Figure 4 ijms-22-03465-f004:**
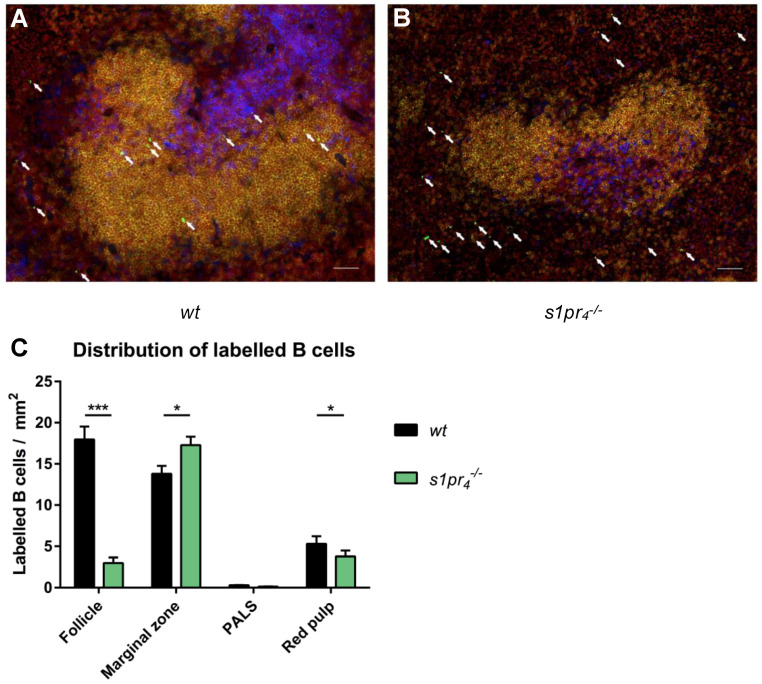
Localization of intraperitoneal transferred peritoneal B cells within the spleen after LPS stimulation in vivo. The localization of *s1pr_4_^−/−^* (**A**) and *wt* (**B**) i.p. transferred peritoneal B cells within the spleen was analyzed using fluorescence microscopy. Representative immunofluorescence-stained splenic sections from LPS-stimulated *wt* mice after adoptive cell transfer of *wt* or *s1pr_4_^−/−^* peritoneal cells (white bar equals 50 µm). The signal intensity of CFSE from labeled cells was enhanced using polyclonal anti-FITC-Alexa488 (green). Tissue sections were additionally stained with anti-B220 (yellow) and anti-CD4 (blue), while nuclei were stained with Draq5 (red). Transferred B cells were identified as B220^+^ CFSE^+^ (white arrows). (**C**) Quantification of the exact localization of transferred B cells within the different splenic and intestinal compartments (*n* = 10). (white scale bar equals 50 µm) * *p* < 0.05, *** *p* < 0.001.

**Figure 5 ijms-22-03465-f005:**
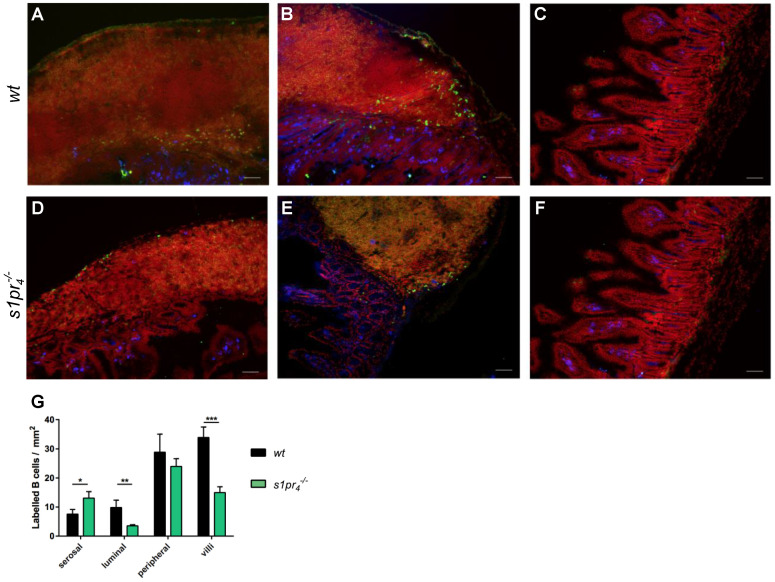
Localization of intraperitoneal transferred peritoneal B cells within Peyer’s patches after LPS stimulation in vivo. The signal intensity of CFSE from labeled cells was enhanced using polyclonal anti-FITC-Alexa488 (green). Tissue sections were additionally stained with anti-B220 (yellow) and anti-IgA (blue), while nuclei were stained with Draq5 (red) (white scale bar equals 50 µm). Transferred B cells were identified as B220^+^ CFSE^+^. While *wt* B cells resided on the luminal side (**A**) of the PPs below the SED, *s1pr_4_^−/−^* cells were found on the serosal side (**D**) of the PP. At the outer edge of the PPs, B cells of both genotypes were found in large numbers (*wt*: (**B**); *s1pr_4_^−/−^*: (**E**)). There were significantly more *wt* (**C**) than *s1pr_4_^−/−^* (**F**) B cells within the intestinal villi. The vast majority of transferred B cells in the small intestine were IgA-negative. (**G**) The number of transferred B cells within the compartments of PPs was quantified using QuPath for the downstream segmentation and classification of cells. * *p* < 0.05, ** *p* < 0.01, *** *p* < 0.001.

**Figure 6 ijms-22-03465-f006:**
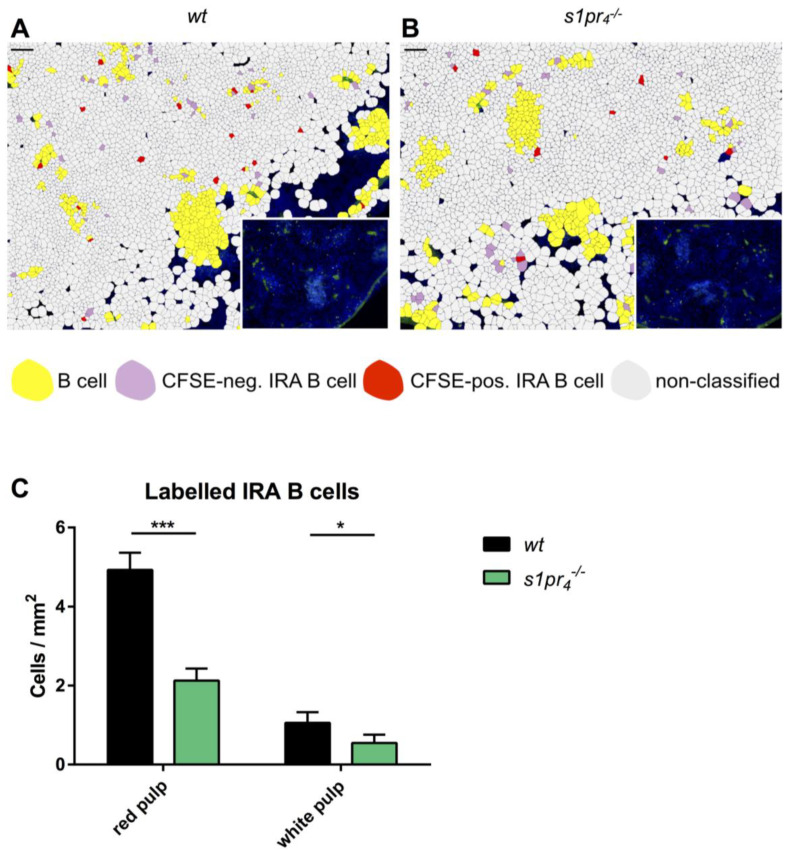
Analysis of the localization of splenic IRA B cells derived from transferred peritoneal B cells. After the adoptive cells of *wt* (**A**) and *s1pr_4_^−/−^* (**B**) peritoneal cells and subsequent LPS stimulation, spleen sections (3 µm) were stained with anti-B220 (green), anti-GM-CSF (red) and anti-FITC (white), while nuclei were stained with DAPI (blue) (black scale bar equals 50 µm). To improve visualization of spatial cell distribution, cell maps of the tissue sections (created using QuPath) are shown, with the original images displayed in the bottom corner. Cells were segmented based on DAPI staining and then classified according to their cytoplasmic staining. “B cells” were identified as B220^+^ CFSE^−^ GM-CSF^−^, “CFSE-neg. IRA B cells” as B220^+^ CFSE^−^ GM-CSF^+^ and “CFSE-pos. IRA B cells” as B220^+^ CFSE^+^ GM-CSF^+^. The remaining cells were designated as “non classified”. (**C**) Based on the aforementioned classification, the number of cells per mm^2^ was calculated and then compared between the genotypes. A total of 10 representative images in two sectional planes per animal were recorded as technical replicates, and their mean value was further used as a single biological replicate. Results shown are the mean values (+SEM) of these biological replicates for *n* = 10 animals per group. * *p* < 0.05, *** *p* < 0.001.

**Figure 7 ijms-22-03465-f007:**
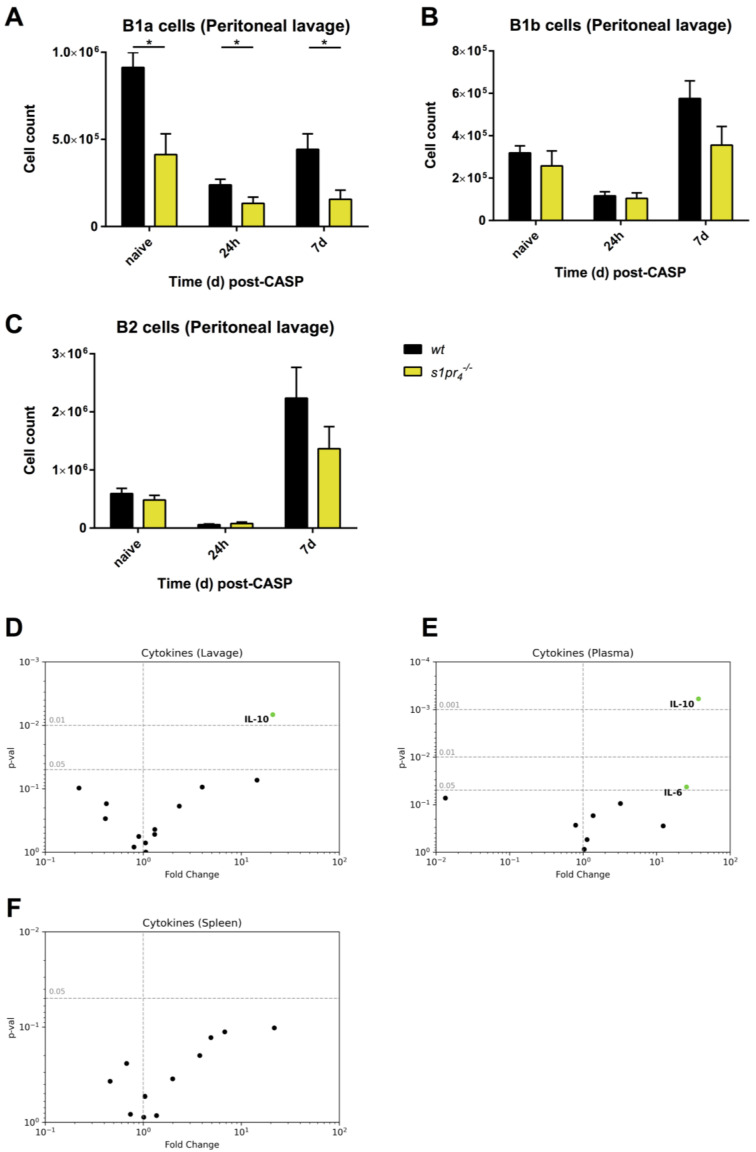
Influence of S1P_4_ expression on the composition of peritoneal B cell populations and cytokine levels in a murine model of polymicrobial abdominal sepsis. Polymicrobial abdominal sepsis was induced by colon ascendens stent peritonitis (CASP) in *wt* and *s1pr_4_^−/−^* mice. (**A**–**C**) The number of B1a (**A**), B1b (**B**)**,** and B2 (**C**) B cells within the peritoneal lavage fluid of naive mice, 24 h and 7 days after induction of CASP. (**D**−**F**) Measurement of cytokines in the lavage (**D**), plasma (**E**), and spleen (**F**) 24 h after the induction of abdominal sepsis. The fold change of concentrations between *wt* and *s1pr_4_^−/−^* animals was calculated and plotted against the corresponding *p*-values (*n* = 10). Only cytokines that reached significance are labeled. * *p* < 0.05.

**Figure 8 ijms-22-03465-f008:**
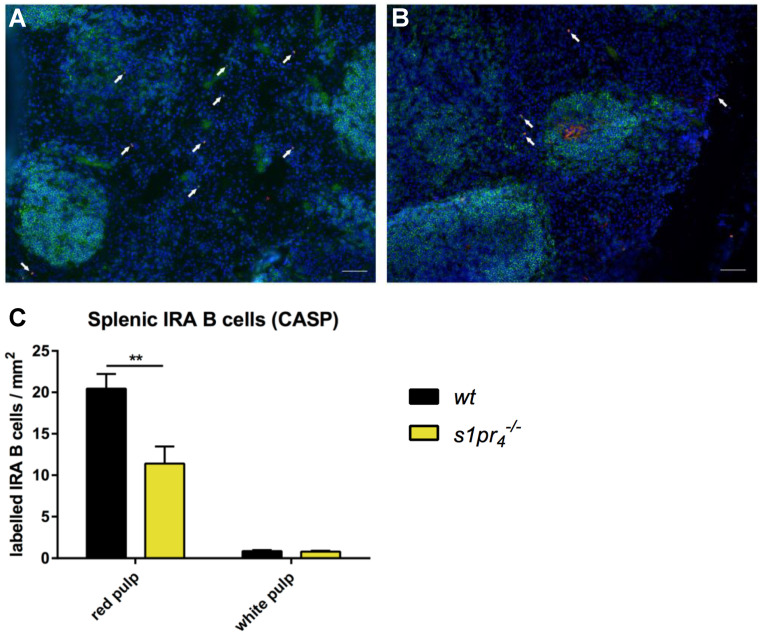
Quantification of IRA B cells in the spleen after CASP. Seven days after sepsis induction by colon ascendens stent peritonitis (CASP), the number and localization of splenic IRA B cells in *wt* (**A**) and *s1pr_4_^−/−^* (**B**) mice were analyzed. Splenic tissue sections were stained with anti-B220 (green), anti-GM-CSF (red), and DAPI (blue). IRA B cells were identified as B220^+^ GM-CSF^+^ cells and are marked with white arrows (white bars equal 50 µm). (**C**) The number of IRA B cells was compared between *wt* and *s1pr_4_^−/−^* animals for both splenic compartments. Bars depict the mean values (+SEM) of IRA B cell counts (*n* = 6) (white scale bar equals 50 µm). ** *p* < 0.01.

## Data Availability

The data presented in this study are available on request from the corresponding author.

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
