# Peer review of "Sphingosine-1-Phosphate Receptor Type 4 (S1P4) Is Differentially Regulated in Peritoneal B1 B Cells upon TLR4 Stimulation and Facilitates the Egress of Peritoneal B1a B Cells and Subsequent Accumulation of Splenic IRA B Cells under Inflammatory Conditions"

_ijms, 2021, doi:10.3390/ijms22073465_

Round 1

Reviewer 1 Report

In their manuscript, Riese et al. investigated the effect of TRL4 stimulation on S1P receptor expression by peritoneal B-cell subpopulations and characterized the functional consequences of S1PR4 deficiency. Initial in-vitro experimentation pointed to a differential downregulation of S1PR4 in B1 B-cells, with the B1 A subtype showing an S1PR4-dependent transmigration defect regardless of TRL4 stimulation. In an adoptive transfer model employing an S1PR4 knockout mouse the authors showed a role for S1PR4 for entry of peritoneal B-cells into perifollicular regions of the spleen and into Peyer's Patches. Similar results were found in a nonsterile peritonitis model concerning the spleen, however, results concerning the emigration from the peritoneum differed.

The fundamental problem I have with this manuscript is that it attempts to link an LPS-induced downregulation of S1PR4 receptor and a role of this receptor in response to a septic event mediated by gram-negative bacteria. A receptor that is downregulated and cells that exhibit much reduced functionality upon an infection can hardly been seen as an important means to ward off this very infection. Yes, it may play a role in the absence of infection, but certainly not as a response element in the infection. The whole concept of this study appears therefore to be blurred and I don't see how it could become clear to the reader without completely redesigning it.

Author Response

Thank you for your critical remarks on our manuscript. We realized that our reasoning has not yet been sufficiently pertinent to show that the downregulation of a receptor implicated in the regulation of immune cell trafficking may have the potential to modify the immune response to a pathogen. We will therefor detail our reasons to think so in the following paragraphs. We have also modified our discussion in order to elucidate our interpretation of the data provided in the article.

Immune cell migration and the appropriated positioning of immune cells within lymphoid organs and peripheral tissues are a prerequisite for the optimal interaction between these cells which in turn is indispensable for both the maintenance of the immune homeostasis as well as for the generation of an adequate immune response in diseased states. These migrational processes are frequently the resultant of the signalling of more than one messenger molecule acting on the cell. In immune cells, migratory processes are essentially controlled by chemokines and sphingosine 1-phospate and their interaction with the respective receptors [1,2]. This system exerts its function not only by tightly regulated levels of the respective ligands, but also by the differential expression of the chemokine and S1P receptors both under healthy and diseased conditions. Examples for the downregulation of membrane bound GPCRs as mechanism for determining cell migration and positioning of immune cells in the immune homeostasis are the chemokine receptors CCR9, CXCR4, CXCR5 as well as the sphingosine 1-phosphate receptor S1P1. Downregulation of CCR9 in memory Treg cells determines the reduced tropism of these cells for the intestine and facilitated the homing of these cells to other peripheral target tissue [2,3]. CXCR4 is downregulated during the maturation of neutrophils in bone marrow, finally promoting the egress of these cells into the blood [4]. CXCR5 is downregulated during the migration of B cells through various compartments of the lymph node [5]. Finally, differential regulation of the S1P1 receptor is primordial for T cell trafficking through secondary lymphoid organs and B cell positioning within the marginal zone [6,7]. In polymicrobial sepsis, CXCR2 downregulation has been shown to reduce neutrophil migration to the peritoneal cavity, resulting in significant changes in sepsis outcome [8].

 All these examples illustrate that receptor downregulation may be a response element in the reaction to an infectious challenge by affecting the balance of various chemotactically active messengers, resulting in a modification of the resulting migrational behaviour of the cells. They show convincingly that at least as far as regulation of migrational processes is concerned, downregulation of membrane bound receptors is a common mechanism to control cell migrations both under homeostatic as well as under pathological conditions.

The data presented in our manuscript do not show reduced functionality of LPS stimulated peritoneal B1 B cells in general. We chose to assess in vitro migration to an S1P gradient in order to test the functional relevance of S1P downregulation. In vivo, this modification of S1P induced chemotactic signal may result in increased responsiveness to other chemotactic stimuli as seen for example in splenic marginal B cells [6].

In order to assess the role of S1P4 signalling (and not the role of LPS-induced S1P4 downregulation, since in this case a constitutively S1P4 expressing model would have been more appropriate), we performed the experiments reported under the headlines 2.3. – 2.7., that showed in our eyes convincingly a role of S1P4 in the regulation of peritoneal B1 B cell trafficking under septic condition.  Among others, cells with prognostic significance for sepsis outcome have been affected by the S1P4 deficiency [9]. Whether these effects of S1P4 deficiency will be sufficient to affect the capacity to ward off the infection will be the subject of further experiments.

We hope that we have been able to clarify the basis of our interpretation of our results. We have modified parts of our discussion to deduce the interpretation of our results more clearly and precisely.

Bibliography:

  1. Cyster JG. Chemokines and cell migration in secondary lymphoid organs. Science 1999; 286(5447): 2098-2102.
  2. Griffith JW, Sokol CL, Luster AD. Chemokines and chemokine receptors: positioning cells for host defense and immunity. Annu Rev Immunol 2014; 32: 659-702.
  3. Lim HW, Broxmeyer HE, Kim CH. Regulation of trafficking receptor expression in human forkhead box P3+ regulatory T cells. J Immunol. 2006; 177(2): 840-851. doi: 810.4049/jimmunol.4177.4042.4840.
  4. Suratt BT, Petty JM, Young SK, Malcolm KC, Lieber JG, Nick JA, Gonzalo JA, Henson PM, Worthen GS. Role of the CXCR4/SDF-1 chemokine axis in circulating neutrophil homeostasis. Blood. 2004; 104(2): 565-571. doi: 510.1182/blood-2003-1110-3638. Epub 2004 Mar 1130.
  5. Park SM, Brooks AE, Chen CJ, Sheppard HM, Loef EJ, McIntosh JD, Angel CE, Mansell CJ, Bartlett A, Cebon J, Birch NP, Dunbar PR. Migratory cues controlling B-lymphocyte trafficking in human lymph nodes. Immunol Cell Biol 2021; 99(1): 49-64.
  6. Cinamon G, Matloubian M, Lesneski MJ, Xu Y, Low C, Lu T, Proia RL, Cyster JG. Sphingosine 1-phosphate receptor 1 promotes B cell localization in the splenic marginal zone. Nat Immunol 2004; 5(7): 713-720.
  7. Lo CG, Xu Y, Proia RL, Cyster JG. Cyclical modulation of sphingosine-1-phosphate receptor 1 surface expression during lymphocyte recirculation and relationship to lymphoid organ transit. J Exp Med 2005; 201(2): 291-301.
  8. Alves-Filho JC, Freitas A, Souto FO, Spiller F, Paula-Neto H, Silva JS, Gazzinelli RT, Teixeira MM, Ferreira SH, Cunha FQ. Regulation of chemokine receptor by Toll-like receptor 2 is critical to neutrophil migration and resistance to polymicrobial sepsis. Proc Natl Acad Sci U S A. 2009; 106(10): 4018-4023. doi: 4010.1073/pnas.0900196106. Epub 0900192009 Feb 0900196120.
  9. Rauch PJ, Chudnovskiy A, Robbins CS, Weber GF, Etzrodt M, Hilgendorf I, Tiglao E, Figueiredo JL, Iwamoto Y, Theurl I, Gorbatov R, Waring MT, Chicoine AT, Mouded M, Pittet MJ, Nahrendorf M, Weissleder R, Swirski FK. Innate response activator B cells protect against microbial sepsis. Science 2012; 335(6068): 597-601.

Reviewer 2 Report

In this manuscript, Riese et al. present findings that relate LPS-induced downregulation of S1P4 receptor to reduction in B1b B cell migration. The authors demonstrate that LPS stimulation of isolated B1a, B1b and B2 cells results in a reduction in S1P1 and S1P4 expression. The authors then demonstrate that LPS lowers chemotaxis of B1a, B1b and B2 cells towards S1P. By using cells harvested from an S1P4 knock-out mouse, the authors were able to show that S1P4 partially contributes to migration of B1a cells, but not B1b or B2 cells. The authors tracked labeled WT or S1P4 KO B cells in vivo, and discovered that KO cells are defective for migration away from the peritoneum, and transfer to the spleen and mesenterial lymph node. The KO cells that are able to migrate to the spleen are restricted to the marginal zone, and excluded from the follicle. The overall effect of this is a reduction of IRA B cell localization to red pulp. The authors use the CASP model to further demonstrate that B1b cell migration is partially dependent upon S1P4 expression. Interestingly, S1P4 knock out mice demonstrated a differential release of cytokines, including an increase in IL-10 in the lavage, as well as IL-10 and IL-6 in plasma. No significant differences were observed in the spleen. Following CASP, IRA B cell migration mirrored the authors' observations from sterile LPS stimulation, demonstrating that S1P4 expression is critical for IRA B cell localization to red pulp. Overall the authors show that LPS can stimulate the downregulation of S1P4, and that reduction of S1P4 expression correlates with a decredase in the migration of B1a B cells to the spleen in response to infection. 

This paper requires minor revisions and figure corrections in order to be suitable for publication: 

Figure 1: The authors claim that the resulting decrease in S1P receptor expression in response to LPS is due to TLR-4 signaling. While LPS has been shown to activate TLR-4, this is information is not outlined or cited in the introduction. The authors do not show in Figure 1 that TLR-4 is activated, nor do they demonstrate that TLR-4 is required for down-regulation of S1P1 or S1P4. The authors should clarify, either through citing previous literature or by demonstrating in this manuscript that TLR-4 is required for LPS-induced S1P receptor downregulation. 

Figure 3. The green and black bars are not labeled or defined in the figure legend. In contrast, Figures 4, 5 and 6 are labeled. 

Minor correction: 

Line 281 - "technics" should be "techniques"

Author Response

Thank you for your helpful comments on our manuscript.

In the introduction, we provide bibliographic evidence that LPS induced B cell activation, which was used as an in vitro model in our experimental setting, is mediated by TLR4 signalling [1].

We have also modified Figure 3 according to your suggestions and corrected the spelling of Techniques in line 281.

Bibliography

1. Minguet S, Dopfer EP, Pollmer C, Freudenberg MA, Galanos C, Reth M, Huber M, Schamel WW. Enhanced B-cell activation mediated by TLR4 and BCR crosstalk. Eur J Immunol. 2008; 38(9): 2475-2487. doi: 2410.1002/eji.200738094.